# TRAINING VARIATIONAL AUTO ENCODERS WITH DISCRETE LATENT REPRESENTATIONS USING IMPORTANCE SAMPLING

## ABSTRACT

The Variational Auto Encoder (VAE) is a popular generative latent variable model that is often applied for representation learning. Standard VAEs assume continuous valued latent variables and are trained by maximization of the evidence lower bound (ELBO). Conventional methods obtain a differentiable estimate of the ELBO with reparametrized sampling and optimize it with Stochastic Gradient Descend (SGD). However, this is not possible if we want to train VAEs with discrete valued latent variables, since reparametrized sampling is not possible. Till now, there exist no simple solutions to circumvent this problem. In this paper, we propose an easy method to train VAEs with binary or categorically valued latent representations. Therefore, we use a differentiable estimator for the ELBO which is based on importance sampling. In experiments, we verify the approach and train two different VAEs architectures with Bernoulli and Categorically distributed latent representations on two different benchmark datasets.

## 1 THE VARIATIONAL AUTO ENCODER

The variational auto encoder (VAE) is a generative model which it is trained to approximate the true data generating distribution $p(\mathbf{x})$ of an observed random vector $\mathbf{x}$ from a given training set $\mathcal{D} = \{\mathbf{x}_1, ..., \mathbf{x}_N\}$ (Kingma & Welling (2013); Kingma et al. (2016)). It is an especially suited model if $\mathbf{x}$ is high dimensional or has highly nonlinear dependent elements. Therefore, the VAE is oftenly used for tasks like density estimation, data generation, data interpolation (White (2016)), outlier and anomaly detection (An & Cho (2015); Xu et al. (2018)) or clustering (Jiang et al. (2016); Dilokthanakul et al. (2016)).

As shown in Fig. 1, the VAE is an easy latent variable model, where the observations $\mathbf{x} \sim p(\mathbf{x}|\mathbf{z})$ are dependent on latent variables $\mathbf{z} \sim p(\mathbf{z})$.

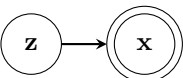

Figure 1: The latent variable model of a VAE with latent variables $\mathbf{z}$ and observations $\mathbf{x}$.

During training, the VAE maximizes the probability $p(\mathbf{x})$ to observe the data $\mathbf{x}$. Therefore, the negative evidence lower bound (ELBO)

$$
\begin{aligned}
\mathcal{L}(\boldsymbol{\theta}) &= -\mathrm{E}_{q(\mathbf{z}|\mathbf{x})}\left[\ln p(\mathbf{x}|\mathbf{z})\right] + \mathrm{D}_{KL}(q(\mathbf{z}|\mathbf{x})||p(\mathbf{z})) & (1)\\
&\geq -\ln p(\mathbf{x}) + \mathrm{D}_{KL}(q(\mathbf{z}|\mathbf{x})||p(\mathbf{z}|\mathbf{x})) & (2)
\end{aligned}
$$

is minimized, where $p(\mathbf{z}|\mathbf{x}) = p(\mathbf{x}|\mathbf{z})p(\mathbf{z})/\int p(\mathbf{x}|\mathbf{z})p(\mathbf{z})\mathrm{d}\mathbf{z}$ is the true but intractable posterior distribution the model assigns to $\mathbf{z}$, $q(\mathbf{z}|\mathbf{x})$ is the corresponding tractable variational approximation and $\mathrm{D}_{KL}(q(\mathbf{z}|\mathbf{x})||p(\mathbf{z}|\mathbf{x}))$ is the Kullback-Leibler (KL) divergence between $p(\mathbf{z}|\mathbf{x})$ and $q(\mathbf{z}|\mathbf{x})$. Because $\mathrm{D}_{KL}(q(\mathbf{z}|\mathbf{x})||p(\mathbf{z}|\mathbf{x})) > 0$, minimizing $\mathcal{L}(\boldsymbol{\theta})$ means to maximize the probability $p(\mathbf{x})$ the model assigns to observations $\mathbf{x}$. Therefore, $\mathrm{D}_{KL}(q(\mathbf{z}|\mathbf{x})||p(\mathbf{z}|\mathbf{x}))$ must be as as close as possible to 0, meaning that after training $q(\mathbf{z}|\mathbf{x})$ is a very good approximation of the true posterior $p(\mathbf{z}|\mathbf{x})$.

Kingma & Welling (2013) proposed to minimize $\mathcal{L}(\boldsymbol{\theta})$, using stochastic gradient descent on a training data set, which they called Stochastic Gradient Variational Bayes (SGVB).

The VAE uses parametric distributions that are parametrized by an encoder network with parameters $\boldsymbol{\theta}_E$ and a decoder network with parameters $\boldsymbol{\theta}_D$ for both $q(\mathbf{z}|\mathbf{x})$ and $p(\mathbf{x}|\mathbf{z})$, respectively. This leads to the well known encoder-decoder structure in Fig. 2. The data likelihood is a distribution with mean $\hat{\mathbf{x}}$, that is the output of the decoder network. Further, we assume in this paper, that the variational posterior $q(\mathbf{z}|\mathbf{x})$ is a distribution from the exponential family

$$q(\mathbf{z}|\mathbf{x}) = \exp(\boldsymbol{\eta}^T(\mathbf{x};\boldsymbol{\theta}_E)T(\mathbf{z}) - A(\boldsymbol{\eta}(\mathbf{x};\boldsymbol{\theta}_E))) \tag{3}$$

with natural parameters $\boldsymbol{\eta}(\mathbf{x};\boldsymbol{\theta}_E)$, sufficient statistic $T(\mathbf{z})$ and log partition function $A(\boldsymbol{\eta}(\mathbf{x};\boldsymbol{\theta}_E))$. This gives us the flexibility to study training with different $q(\mathbf{z}|\mathbf{x})$ in the same mathematical framework. As shown in Fig. 2, the natural parameters $\boldsymbol{\eta}$ are the output of the encoder network, where we drop the arguments $\mathbf{x}, \boldsymbol{\theta}_E$ for shorter notations in the remainder of the paper.

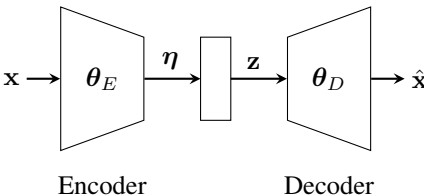

Encoder          Decoder

Figure 2: The encoder-decoder structure of a VAE. The encoder parametrizes $q(\mathbf{z}|\mathbf{x})$ as an exponential family distribution with natural parameters $\boldsymbol{\eta}$ and the decoder parametrizes $p(\mathbf{x}|\mathbf{z})$ with mean $\hat{\mathbf{x}}$.

The conventional VAE proposed in (Kingma & Welling (2013); Kingma et al. (2016)) learns continuous latent representations $\mathbf{z} \in \mathbb{R}^c$. It uses i.i.d. Gaussian distributed $\mathbf{z}$, meaning that $\boldsymbol{\eta} = [\mu_1/\sigma_1^2, -1/(2\sigma_1^2), ..., \mu_c/\sigma_c^2, -1/(2\sigma_c^2)]^T$, $T(\mathbf{z}) = [z_1, z_1^2, ..., z_c, z_c^2]^T$ and $A(\boldsymbol{\eta})$ is chosen such that $q(\mathbf{z}|\mathbf{x})$ integrates to one. The likelihood is also Gaussian, with $p(\mathbf{x}|\mathbf{z}) \sim N(\hat{x}, \mathbf{1})$. But in many applications learning discrete rather than continuous representations is advantageous. Binary representations $\mathbf{z} \in \{0,1\}^c$ can for example be used very efficiently for hashing, what is a powerful method for large-scale visual search (Liong et al. (2015)). Learning Categorical representations $\mathbf{z} \in \{\mathbf{e}_1, ..., \mathbf{e}_c\}$ is interesting, because this naturally lead to clustering of the data $\mathbf{x}$, as shown in the experiments. Further, for both binary and categorical $\mathbf{z}$ it is easy to find entropy based heuristics to choose the size of the latent space, because the entropy is bounded for discrete $\mathbf{z}$.

However, training VAEs with discrete latent representations is problematic, since standard SGVB can not be applied for optimization. Because SGVB is a gradient based method, we need to calculate the derivative of the two cost terms with respect to the encoder and decoder parameters

$$\frac{\partial}{\partial\boldsymbol{\theta}}\mathcal{L}_{KL}(\boldsymbol{\theta}) = \frac{\partial}{\partial\boldsymbol{\theta}}\mathrm{D}_{KL}(q(\mathbf{z}|\mathbf{x})||p(\mathbf{z})) \tag{4}$$

$$\frac{\partial}{\partial\boldsymbol{\theta}}\mathcal{L}_L(\boldsymbol{\theta}) = \frac{\partial}{\partial\boldsymbol{\theta}}\mathrm{E}_{q(\mathbf{z}|\mathbf{x})}\left[\ln p(\mathbf{x}|\mathbf{z})\right], \tag{5}$$

where $\mathcal{L}_{KL}(\boldsymbol{\theta})$ only depends on the encoder parameters and the expected log likelihood term $\mathcal{L}_L(\boldsymbol{\theta})$ depends on both encoder and decoder parameters. For a suited choice of $p(\mathbf{z})$ and $q(\mathbf{z}|\mathbf{x})$, $\mathcal{L}_{KL}(\boldsymbol{\theta})$ can be calculated in closed form. However, $\mathcal{L}_L(\boldsymbol{\theta})$ contains an expectation over $\mathbf{z} \sim q(\mathbf{z}|\mathbf{x})$ that has to be estimated during training. A good estimator $\hat{\mathcal{L}}_L(\boldsymbol{\theta})$ for $\mathcal{L}_D(\boldsymbol{\theta})$ that is unbiased, differentiable with respect to $\boldsymbol{\theta}$ and that has low variance is the key to train VAEs. SGVB uses an estimator $\hat{\mathcal{L}}_L^R(\boldsymbol{\theta})$ that is based on reparametrization of $q(\mathbf{z}|\mathbf{x})$ and sampling (Kingma & Welling (2013)). However, as described in section 2, this method places many restrictions on the form of $q(\mathbf{z}|\mathbf{x})$ and fails if $q(\mathbf{z}|\mathbf{x})$ can not be reparametrized. This is the case if $\mathbf{z}$ is discrete, for example.

In this paper, we propose a simple and differentiable estimator $\hat{\mathcal{L}}_L^I(\boldsymbol{\theta})$ for $\mathcal{L}_L(\boldsymbol{\theta})$ that is based on importance sampling. Because no reparametrization is needed, it can be used to train VAEs with binary or categorical latent representations. Compared to previously proposed methods like the Vector Quantised-Variational Auto Encoder (VQ-VAE) (van den Oord et al. (2017)), which is based on a straight-through estimator for the gradient of $\mathcal{L}_L(\boldsymbol{\theta})$ (Bengio et al. (2013)), our proposed estimator has two advantages. It is unbiased and its variance approaches zero the closer we are to the optimum.

## 2   ESTIMATING THE EXPECTED LOG-LIKELIHOOD WITH REPARAMETRIZED SAMPLING

The standard estimator $\hat{\mathcal{L}}_L^R(\boldsymbol{\theta})$ proposed in Kingma & Welling (2013) is based on reparametrized sampling

$$
\begin{aligned}
\frac{\partial}{\partial \boldsymbol{\theta}} \mathcal{L}_L(\boldsymbol{\theta}) &= \frac{\partial}{\partial \boldsymbol{\theta}} \mathrm{E}_{q(\mathbf{z}|\mathbf{x})} \left[ \ln p(\mathbf{x}|\mathbf{z}) \right] & (6) \\
&= \frac{\partial}{\partial \boldsymbol{\theta}} \mathrm{E}_{p(\epsilon)} \left[ \ln p(\mathbf{x}|\mathbf{z} = f(\epsilon, \boldsymbol{\theta})) \right] & (7) \\
&\approx \frac{\partial}{\partial \boldsymbol{\theta}} \frac{1}{M} \sum_{m=1}^{M} \ln p(\mathbf{x}|\mathbf{z} = f(\epsilon_m, \boldsymbol{\theta})) & (8) \\
&= \frac{\partial}{\partial \boldsymbol{\theta}} \hat{\mathcal{L}}_L^R(\boldsymbol{\theta}) & (9)
\end{aligned}
$$

where $\epsilon$ is a random variable with the distribution $p(\epsilon)$, $\epsilon_m$ are samples from this distribution and $f(\epsilon, \boldsymbol{\theta})$ is a reparametrization function, such that $\mathbf{z} = f(\epsilon, \boldsymbol{\theta}) \sim q(\mathbf{z}|\mathbf{x})$. This estimator can be used to train VAEs with SGVB if two conditions are fulfilled: I) There exists a distribution $p(\boldsymbol{\epsilon})$ and a reparametrization function $f(\epsilon, \boldsymbol{\theta})$, such that $\mathbf{z} = f(\epsilon, \boldsymbol{\theta}) \sim q(\mathbf{z}|\mathbf{x})$. II) The derivative of Eq. 5 must exist. With Eq. 8, we obtain

$$
\frac{\partial}{\partial \boldsymbol{\theta}} \hat{\mathcal{L}}_L^R(\boldsymbol{\theta}) = \frac{1}{M} \sum_{m=1}^{M} \frac{\partial}{\partial \mathbf{z}} \ln p(\mathbf{x}|\mathbf{z} = f(\epsilon_m, \boldsymbol{\theta})) \frac{\partial}{\partial \boldsymbol{\theta}} \mathbf{z}, \tag{10}
$$

meaning that both the reparametrization function $f(\epsilon, \boldsymbol{\theta})$ and $\ln p(\mathbf{x}|\mathbf{z})$ must be differentiable with respect to $\mathbf{z}$ and $\boldsymbol{\theta}$, respectively, to allow direct backpropagation of the gradient through the reparametrized sampling operator. If these conditions are fulfilled, the gradient can flow directly from the output to the input layer of the VAE, as shown in Fig. 3. Distributions over discrete latent representations $\mathbf{z}$ can not be reparametrized this way. Therefore, this estimator can not be used to train VAEs with such representations.

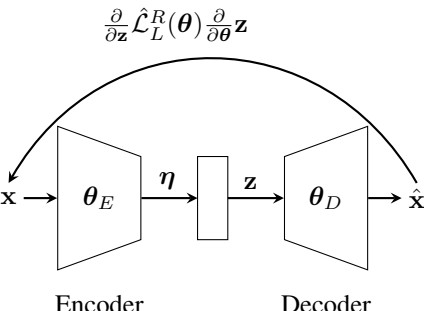

Figure 3: The gradient flow through the VAE, using $\hat{\mathcal{L}}_L^R(\boldsymbol{\theta})$ based on reparametrized sampling. The gradient is propagated directly through the reparametrized sampling operator.

## 3 ESTIMATING THE EXPECTED LOG-LIKELIHOOD WITH IMPORTANCE SAMPLING

We propose an estimator $\hat{\mathcal{L}}_L^I(\boldsymbol{\theta})$ which is based on importance sampling and can also be used to train VAEs with binary or categorical latent representations $\mathbf{z}$. Expanding Eq. 4 leads to

$$\frac{\partial}{\partial\boldsymbol{\theta}}\mathcal{L}_L(\boldsymbol{\theta}) = \frac{\partial}{\partial\boldsymbol{\theta}}\int \ln p(\mathbf{x}|\mathbf{z})q(\mathbf{z}|\mathbf{x})\mathrm{d}\mathbf{z} \tag{11}$$

$$= \frac{\partial}{\partial\boldsymbol{\theta}}\int \ln p(\mathbf{x}|\mathbf{z})\frac{q(\mathbf{z}|\mathbf{x})}{q_I(\mathbf{z})}q_I(\mathbf{z})\mathrm{d}\mathbf{z} \tag{12}$$

$$\approx \frac{\partial}{\partial\boldsymbol{\theta}}\frac{1}{M}\left(\sum_{m=1}^M \ln p(\mathbf{x}|\mathbf{z}_m)\frac{q(\mathbf{z}_m|\mathbf{x})}{q_I(\mathbf{z}_m)}\right) \tag{13}$$

$$= \frac{\partial}{\partial\boldsymbol{\theta}}\hat{\mathcal{L}}_L^I(\boldsymbol{\theta}), \tag{14}$$

where $q_I(\mathbf{z})$ is an arbitrary distribution that is of the same form as $q(\mathbf{z}|\mathbf{x})$ which is independent from the parameters $\boldsymbol{\theta}$. $\mathbf{z}_m \sim q_I(\mathbf{z})$ are samples from this distribution. The estimator computes a weighted sum of the log likelihood $\ln p(\mathbf{x}|\mathbf{z}_m)$ with the weighting $q(\mathbf{z}_m|\mathbf{x})/q_I(\mathbf{z})$.

The benefit is that the log likelihood $\ln p(\mathbf{x}|\mathbf{z}_m)$ depends on the decoder parameters $\boldsymbol{\theta}_D$ only and not on the encoder parameters $\boldsymbol{\theta}_E$ whereas the weighting $q(\mathbf{z}_m|\mathbf{x})/q_I(\mathbf{z})$ depends only on $\boldsymbol{\theta}_D$ and not on $\boldsymbol{\theta}_E$. Therefore, calculation of the gradient of $\hat{\mathcal{L}}_L^I(\boldsymbol{\theta})$ can be separated

$$\frac{\partial}{\partial\boldsymbol{\theta}}\hat{\mathcal{L}}_L^I(\boldsymbol{\theta}) = \left[\frac{\partial}{\partial\boldsymbol{\theta}_E}\hat{\mathcal{L}}_L^I(\boldsymbol{\theta}), \frac{\partial}{\partial\boldsymbol{\theta}_D}\hat{\mathcal{L}}_L^I(\boldsymbol{\theta})\right], \tag{15}$$

with

$$\frac{\partial}{\partial\boldsymbol{\theta}_E}\hat{\mathcal{L}}_L^I(\boldsymbol{\theta}) = \frac{1}{M}\sum_{m=1}^M \ln p(\mathbf{x}|\mathbf{z}_m)\frac{\partial}{\partial\boldsymbol{\theta}_E}\frac{q(\mathbf{z}_m|\mathbf{x})}{q_I(\mathbf{z}_m)} \tag{16}$$

$$\frac{\partial}{\partial\boldsymbol{\theta}_D}\hat{\mathcal{L}}_L^I(\boldsymbol{\theta}) = \frac{1}{M}\sum_{m=1}^M \frac{q(\mathbf{z}_m|\mathbf{x})}{q_I(\mathbf{z}_m)}\frac{\partial}{\partial\boldsymbol{\theta}_D}\ln p(\mathbf{x}|\mathbf{z}_m). \tag{17}$$

As shown in Fig. 4, gradient backpropagation is split into two separate parts. $\frac{\partial}{\partial\boldsymbol{\theta}_D}\hat{\mathcal{L}}_L^I(\boldsymbol{\theta})$ back-propagates the error $\ln p(\mathbf{x}|\mathbf{z})$ from the output of the VAE to the sampling operator and $\frac{\partial}{\partial\boldsymbol{\theta}_E}\hat{\mathcal{L}}_L^I(\boldsymbol{\theta})$ backpropagates the error $\frac{q(\mathbf{z}|\mathbf{x})}{q_I(\mathbf{z})}$ from the sampling operator to the input layer of the VAE.

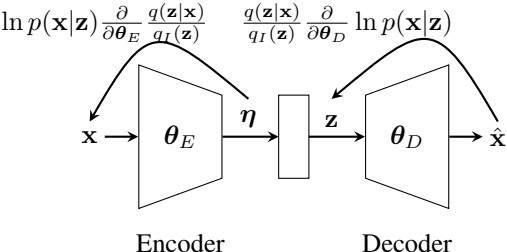

Figure 4: Gradient flow through the VAE when using $\hat{\mathcal{L}}_L^I(\boldsymbol{\theta})$, based on importance sampling.

Compared to $\hat{\mathcal{L}}_L^R(\boldsymbol{\theta})$, we do not need to find a differentiable reparametrization for $q(\mathbf{z}|\mathbf{x})$, because we do not propagate the gradient through the sampling operator. Therefore, $\hat{\mathcal{L}}_L^I(\boldsymbol{\theta})$ can also be used if no reparametrization function exists for $q(\mathbf{z}|\mathbf{x})$, e.g. if it is a Bernoulli or a Categorical distribution.

## 4 VAE WITH BERNOULLI DISTRIBUTED z (BVAE)

Assume the latent representation $\mathbf{z}$ has i.i.d. components that are Bernoulli distributed, i.e. both the variational posterior distribution $q(\mathbf{z}|\mathbf{x})$ and $q_I(\mathbf{z})$ have the form

$$q(\mathbf{z}|\mathbf{x}) = \exp(\boldsymbol{\eta}^T\mathbf{z} - A(\boldsymbol{\eta})) \tag{18}$$

$$q_I(\mathbf{z}) = \exp(\boldsymbol{\xi}^T\mathbf{z} - A(\boldsymbol{\xi})), \tag{19}$$

where $\mathbf{z} \in \{0,1\}^c$, $\boldsymbol{\eta} = [\ln(q_1/(1-q_1)), ..., \ln(q_c/(1-q_c))]$ is the output vector of the encoder that contains the logits of the independent Bernoulli distributions and $A(\boldsymbol{\eta}) = \mathbf{1}^T \ln(1+e^{\boldsymbol{\eta}})$ are the corresponding log-partition functions.

Hence, Eq. 17 is

$$
\begin{aligned}
\frac{\partial}{\partial\boldsymbol{\theta}_E}\hat{\mathcal{L}}_L^I(\boldsymbol{\theta}) &= \frac{1}{M}\sum_{m=1}^M \ln p(\mathbf{x}|\mathbf{z}_m)\frac{\partial}{\partial\boldsymbol{\eta}}\exp\left((\boldsymbol{\eta}-\boldsymbol{\xi})^T\mathbf{z}_m - (A(\boldsymbol{\eta})-A(\boldsymbol{\xi}))\right)\left(\frac{\partial}{\partial\boldsymbol{\theta}_E}\boldsymbol{\eta}\right) \\
&= \frac{1}{M}\sum_{m=1}^M \ln p(\mathbf{x}|\mathbf{z}_m)\exp\left((\boldsymbol{\eta}-\boldsymbol{\xi})^T\mathbf{z}_m - (A(\boldsymbol{\eta})-A(\boldsymbol{\xi}))\right)(\mathbf{z}-\mathbf{q})^T\left(\frac{\partial}{\partial\boldsymbol{\theta}_E}\boldsymbol{\eta}\right),
\end{aligned}
$$

where $\mathbf{q} = [q_1,...,q_c]^T = \frac{1}{1+e^{-\boldsymbol{\eta}}}$ contains the probabilities $q(z_i = 1|\mathbf{x})$. The variance of the estimator $\hat{\mathcal{L}}_L^I(\boldsymbol{\theta})$ heavily depends on the choice of the natural parameters $\boldsymbol{\xi}$ of the distribution $q_I(\mathbf{z})$. We choose $\boldsymbol{\xi} = \boldsymbol{\eta}$, leading to a gradient of the very simple form

$$\frac{\partial}{\partial\boldsymbol{\theta}_E}\hat{\mathcal{L}}_L^I(\boldsymbol{\theta}) = \frac{1}{M}\sum_{m=1}^M \ln p(\mathbf{x}|\mathbf{z}_m)(\mathbf{z}_m-\mathbf{q})^T\left(\frac{\partial}{\partial\boldsymbol{\theta}_E}\boldsymbol{\eta}\right). \tag{20}$$

This estimator of the gradient has two desirable properties for training, which can be easily seen in the one dimensional case with $z \in \{0,1\}$. The mean of the estimator is

$$
\begin{aligned}
\mathrm{E}_{q_I(z)}\left[\frac{\partial}{\partial\boldsymbol{\theta}_E}\hat{\mathcal{L}}_L^I(\boldsymbol{\theta})\right] &= \mathrm{E}_{q_I(z)}\left[\frac{1}{M}\sum_{m=1}^M \ln p(\mathbf{x}|z)(z_m-q)\left(\frac{\partial}{\partial\boldsymbol{\theta}_E}\eta\right)\right] \tag{21}\\
&= q(1-q)(\ln p(\mathbf{x}|z=1) - \ln(p(\mathbf{x}|z=0))\left(\frac{\partial}{\partial\boldsymbol{\theta}_E}\eta\right) \tag{22}\\
&= \frac{\partial}{\partial\boldsymbol{\theta}_E}\mathcal{L}_L(\boldsymbol{\theta}), \tag{23}
\end{aligned}
$$

meaning that the estimator is unbiased. Further, the variance of $\frac{\partial}{\partial\boldsymbol{\theta}_E}\hat{\mathcal{L}}_L^I(\boldsymbol{\theta})$ reduces to zero, the closer $q$ is to 0 or 1, because $q(1-q) \to 0$ and hence the variance of the estimator approaches 0.

That is desirable, since there are only three interesting cases during training that are shown in Fig. 5:

(a) $\ln p(\mathbf{x}|z = 0) = \ln p(\mathbf{x}|z = 1)$: In this case the expected log likelihood $\mathrm{E}_{q(\mathbf{z}|\mathbf{x})}[\ln p(\mathbf{x}|z=0)] = \ln p(\mathbf{x}|z=0)$ we want to maximize is independent of the choice of $q$.

(b) $\ln p(\mathbf{x}|z = 0) > \ln p(\mathbf{x}|z = 1)$: In this case the expected log likelihood $\mathrm{E}_{q(\mathbf{z}|\mathbf{x})}[\ln p(\mathbf{x}|z=0)] = (1-q)\ln p(\mathbf{x}|z=0) + q\ln p(\mathbf{x}|z=1)$ is maximized for $q = 0$.

(c) $\ln p(\mathbf{x}|z = 0) < \ln p(\mathbf{x}|z = 1)$: In this case the expected log likelihood $\mathrm{E}_{q(\mathbf{z}|\mathbf{x})}[\ln p(\mathbf{x}|z=0)] = (1-q)\ln p(\mathbf{x}|z=0) + q\ln p(\mathbf{x}|z=1)$ is maximized for $q = 1$.

This means, that the only candidate points that maximize the log likelihood are $q = 0$ or $q = 1$ lie near $q(z = 1|\mathbf{x}) = 0/1$. Therefore, the longer we train, the more accurate the gradient estimate will be.

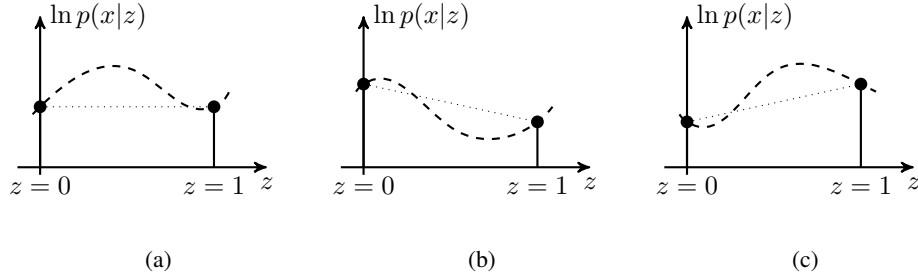

(a)  (b)  (c)

Figure 5: Three different cases how the log-likelihood can vary over $z$. The decoder network defines $\ln(\mathbf{x}|z)$ over $z \in \mathbb{R}$. However, only the points $z = 0$ and $z = 1$ are of interest. (a) $\ln(\mathbf{x}|z = 0) = \ln(\mathbf{x}|z = 1)$, meaning $\mathrm{E}_{q(z|\mathbf{x})}[\ln(\mathbf{x}|z)]$ is independent of $q$. (b) $\ln(\mathbf{x}|z = 0) > \ln(\mathbf{x}|z = 1)$, meaning $\mathrm{E}_{q(z|\mathbf{x})}[\ln(\mathbf{x}|z)]$ is maximized for $q = 0$. (c) $\ln(\mathbf{x}|z = 0) < \ln(\mathbf{x}|z = 1)$, meaning $\mathrm{E}_{q(z|\mathbf{x})}[\ln(\mathbf{x}|z)]$ is maximized for $q = 1$.

## 5  VAE WITH CATEGORICALLY DISTRIBUTED $\mathbf{z}$ (CVAE)

For Categorically distributed $\mathbf{z}$ both the variational posterior distribution $q(\mathbf{z}|\mathbf{x})$ and $q_I(\mathbf{z})$, again have the form

$$q(\mathbf{z}|\mathbf{x}) = \exp(\boldsymbol{\eta}^T \mathbf{z} - A(\boldsymbol{\eta})) \tag{24}$$

$$q_I(\mathbf{z}) = \exp(\boldsymbol{\xi}^T \mathbf{z} - A(\boldsymbol{\xi})), \tag{25}$$

but now $\mathbf{z} \in \{\mathbf{e}_1, ..., \mathbf{e}_c\}$ can assume only $c$ different values. The vector of natural parameters is $\boldsymbol{\eta} = [\ln(p_1/p_c), ..., \ln(p_{c-1}/p_c), 0)]$ and the log partition function is $A(\boldsymbol{\eta}) = \ln \mathbf{1}^T e^{\boldsymbol{\eta}}$.

With the formulas above, we arrive at the same easy form of the expected gradient of the log likelihood

$$\frac{\partial}{\partial \boldsymbol{\theta}_E} \hat{\mathcal{L}}_L^I(\boldsymbol{\theta}) = \frac{1}{M} \sum_{m=1}^{M} \ln p(\mathbf{x}|\mathbf{z}_m) (\mathbf{z}_m - \mathbf{q})^T \left( \frac{\partial}{\partial \boldsymbol{\theta}_E} \boldsymbol{\eta} \right), \tag{26}$$

but now with $\mathbf{q} = \mathrm{softmax}(\boldsymbol{\eta})$ that consists of the probabilities $q_i = q(\mathbf{z} = \mathbf{e}_i)$, where $\sum_{i=1}^{c} q_i = 1$.

## 6  EXPERIMENTS

In the following section, we show our preliminary experiments on the MNIST and Fashion MNIST datasets LeCun & Cortes (2010); Xiao et al. (2017). Two different kinds of VAEs have been evaluated:

1. The BVAE with Bernoulli distributed $\mathbf{z} \in \{0, 1\}^c$.
2. The CVAE with Categorically distributed $\mathbf{z} \in \{\mathbf{e}_1, ..., \mathbf{e}_c\}$.

To train both architectures, the estimator $\hat{\mathcal{L}}_L^I(\boldsymbol{\theta})$ derived in Sec. 5 is used.

Both BVAE and CVAE are tested with two different architectures given in Tab. 1. The fully connected architecture has 2 dense encoder and decoder layers. The encoder and decoder networks of the convolutional architecture consist of 4 convolutional layers and one dense layer each.

In our first experiment we train a FC_BVAE with $c = 50$, i.e. $\mathbf{z} \in \{0, 1\}^{50}$ and a FC_CVAE with $c = 100$, i.e. $\mathbf{z} \in \{\mathbf{z}_1, ..., \mathbf{z}_{100}\}$. We train them for 300 epochs on the MNIST dataset, using

Table 1: The architectural details of the trained VAEs. FC, $\mathrm{Conv}$ and $\mathrm{Conv}^{-1}$ are the fully connected, convolutional and deconvolutional layer, respectively. The shape of the convolutional layers is given in the form $\mathrm{DimA} \times \mathrm{DimB} \times \mathrm{Channel/Stride/Activation}$.

| Architecture | In/Out | Encoder | Latent | Decoder |
|---|---|---|---|---|
| FC_BVAE | 784 | FC 1024/ReLu | $\mathbf{z} \in \mathbb{R}^c \sim \mathrm{Ber}(\frac{1}{1+e^{-\eta}})$ | FC 1024/ReLu |
| or FC_CVAE | | FC $c$/linear | or $\mathbf{z} \in \mathbb{R}^c \sim \mathrm{Cat}(\frac{e^{\eta}}{e^{\mathbf{1}^T \eta}})$ | FC 784/sigmoid |
| CNN_BVAE | 28x28x1 | Conv 3x3x32/2/ReLu | $\mathbf{z} \in \mathbb{R}^c \sim \mathrm{Ber}(\frac{1}{1+e^{-\eta}})$ | FC $c$/ReLu |
| or CNN_CVAE | | Conv 3x3x64/2/ReLu | or $\mathbf{z} \in \mathbb{R}^c \sim \mathrm{Cat}(\frac{e^{\eta}}{e^{\mathbf{1}^T \eta}})$ | reshape |
| | | Conv 3x3x64/2/ReLu | | $\mathrm{Conv}^{-1}$ 3x3x64/2/ReLu |
| | | Conv 3x3x128/2/ReLu | | $\mathrm{Conv}^{-1}$ 3x3x64/2/ReLu |
| | | flatten | | $\mathrm{Conv}^{-1}$ 3x3x32/2/ReLu |
| | | FC $c$/linear | | $\mathrm{Conv}^{-1}$ 3x3x1/2/sigmoid |

SGVB with our proposed estimator $\hat{\mathcal{L}}_L^I(\boldsymbol{\theta})$, to estimate the expected log likelihood, and ADAM as optimizer. Fig. 6 shows the convergence of the loss, the log likelihood the VAEs assign to the training data $\ln p(\mathbf{x}|\mathbf{z})$ and the variance of the estimator $\hat{\mathcal{L}}_L^I(\boldsymbol{\theta})$, for a learning rate of $1e-3$ and a batch size of 2048. During training, the loss decreases steadily without oscillation. We observe that the variance of the estimator $\hat{\mathcal{L}}_L^I(\boldsymbol{\theta})$ decreases the longer we train and the closer we get to the optimum. This is consistent with our theoretical considerations in Sec. 4. The results of the corresponding simulations with the CNN_BVAE and the CNN_CVAE are shown in Fig. 9, in the appendix.

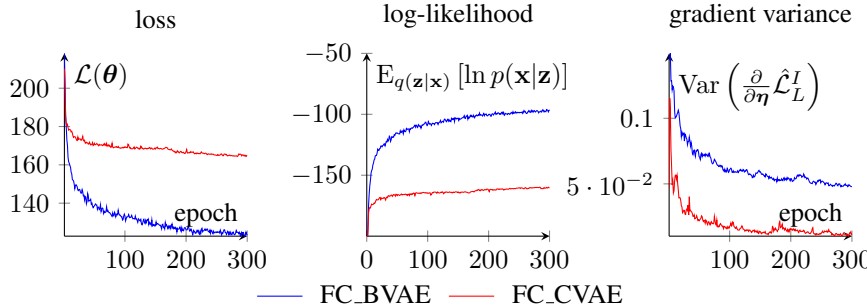

Figure 6: Convergence of the loss, log-likelihood and gradient variance over 300 epochs of training on the MNIST dataset.

The performance of the FC_CVAE is worse than the performance of the FC_BVAE. Training converges to a lower log likelihood $\ln p(\mathbf{x}|\mathbf{z})$, because the maximal information content $H_{CVAE}(\mathbf{z}) \leq \ln(100)$ of the latent variables of the FC_CVAE is much less than the maximal information content $H_{BVAE}(\mathbf{z}) \leq c \ln(2)$ of the latent variables of the FC_BVAE. The FC_CVAE can at maximum learn to generate 100 different handwritten digits, what is a small number compared to the $2^{50}$ different images that the FC_CVAE can learn to generate.

Fig. 7 shows handwritten digits that are generated by the FC_BVAE and the FC_CVAE if we sample $\mathbf{z}$ from the variational posterior $q(\mathbf{z}|\mathbf{x})$. To draw samples from $q(\mathbf{z}|\mathbf{x})$, we feed test data which has not been seen during training to the encoders. The test data is shown in Fig. 7a and Fig. 7c. The corresponding reconstructions generated by the decoders are shown in Fig. 7b and Fig. 7d. Both input and reconstructed images are very similar in case of the FC_BVAE, meaning that it can approximate the data generating distribution $p(\mathbf{x})$ well. However, in case of the FC_CVAE, the generated are blurry and look very different than the input of the encoder. In some cases, the class of the generated digit is even flipped. This happens because of the the very limited model capacity. Similar results for the CNN_BVAE and CNN_CVAE are shown in Fig. 10 in the appendix.

Fig. 8a shows generated images of the FC_BVAE if we sample $\mathbf{z} \sim p(\mathbf{z})$ from the prior distribution. A few generated images look like templates of handwritten digits and the remaining generated images seem to resemble mixtures of different digits. This is similar to the behaviour of a VAE with continuous latent variables, where we can interpolate between or generate mixtures of different

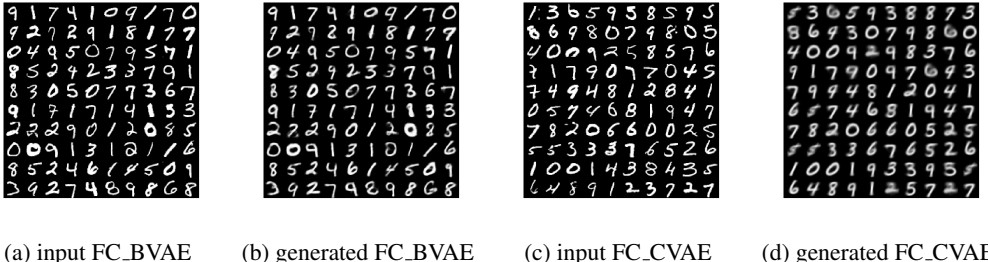

(a) input FC_BVAE      (b) generated FC_BVAE      (c) input FC_CVAE      (d) generated FC_CVAE

Figure 7: Test input images and generated handwritten images of the FC_BVAE and FC_CVAE

digits by traveling through the latent space Kingma & Welling (2013). However, in comparison to VAEs with continuous latent variables, we can only generate discrete mixtures for VAEs if the latent variables $\mathbf{z}$ are Bernoulli.

Fig. 8b shows generated images of the FC_CVAE if we sample $\mathbf{z} \sim p(\mathbf{z})$ from the prior distribution. Since the FC_CVAE can only learn to generate 100 different images, its decoder learns to generate template images that fit well to all the training images. We observe that some latent representations are decoded to meaningless patterns that just fit well to the data in avarage. However, the decoder also learned to generate at least one template image for each class of handwritten digits. Hence, the categorical latent representation can be interpreted as the cluster affiliation and the encoder of the FC_CVAE automatically learns to cluster the data. Similar results for the CNN_BVAE and CNN_CVAE are shown in Fig. 11 in the appendix.

A major drawback of the FC_CVAE is, that the latent space of the FC_CVAE can encode only very little information and thus its generative capabilites are poor. However, we think that they can be increased considerably if we allow a hybrid latent space with some continuous latent variables, as proposed in Chen et al. (2016). This could lead to a powerfull model for nonlinear clustering.

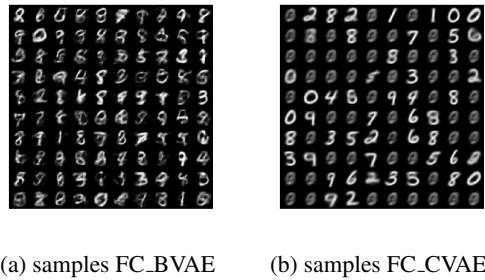

(a) samples FC_BVAE      (b) samples FC_CVAE

Figure 8: Handwritten digits generated by the FC_BVAE and FC_CVAE, if $\mathbf{z}$ is sampled from the prior distribution $p(\mathbf{z})$.

## 7   CONCLUSION

In this paper, we derived an easy estimator for the ELBO, which does not rely on reparametrized sampling and therefore can be used to obtain differentiable estimates, even if reparametrization is not possible, e.g. if the latent variables $\mathbf{z}$ are Bernoulli or Categorically distributed. We have shown theoretically and in experiments, close to the optimal parameter configuration, the variance of the estimator approaches zero. This is a very desirable property for training.

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

## A    RESULTS FOR CNN_BVAE AND CNN_CVAE ON MNIST

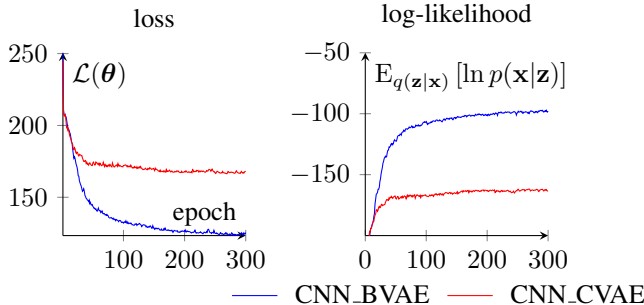

Figure 9: Convergence of the loss and the log-likelihood over 300 epochs, while training on the MNIST dataset.

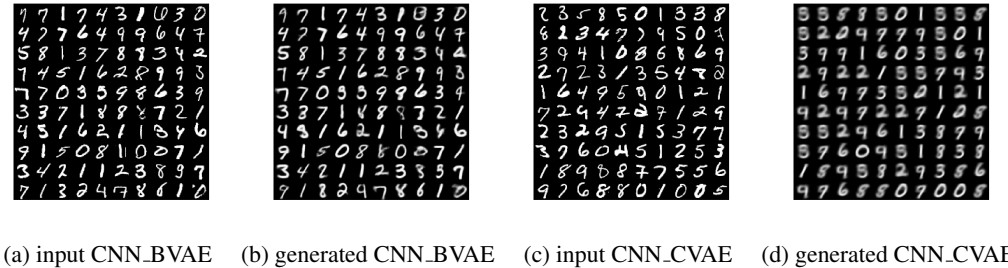

(a) input CNN_BVAE    (b) generated CNN_BVAE    (c) input CNN_CVAE    (d) generated CNN_CVAE

Figure 10: Test input images and generated handwritten images of the CNN_BVAE and CNN_CVAE

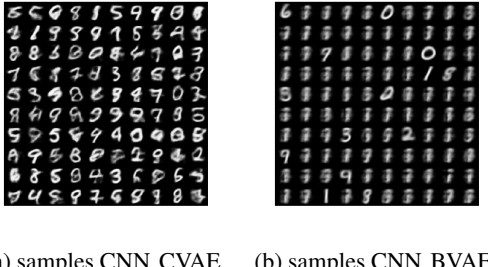

(a) samples CNN_CVAE    (b) samples CNN_BVAE

Figure 11: Handwritten digits generated by the CNN_BVAE and CNN_CVAE, if $\mathbf{z}$ is sampled from the prior distribution $p(\mathbf{z})$.

# B RESULTS FOR CNN_BVAE ON FASHION MNIST

As shown in Fig. 12, the gradient variance approaches 0, the closer we get to the optimum. This is the same behaviour as for the MNIST dataset. As shown in Fig. 13, the FC_BVAE can correctly reconstruct the shape of the given clothes with high accuracy. However, details like texture are lost. This is due to the limited model capacity, i.e. the latent representation $\mathbf{z} \in \{0,1\}^{50}$ of the given VAE can at most encode 50Bits of information.

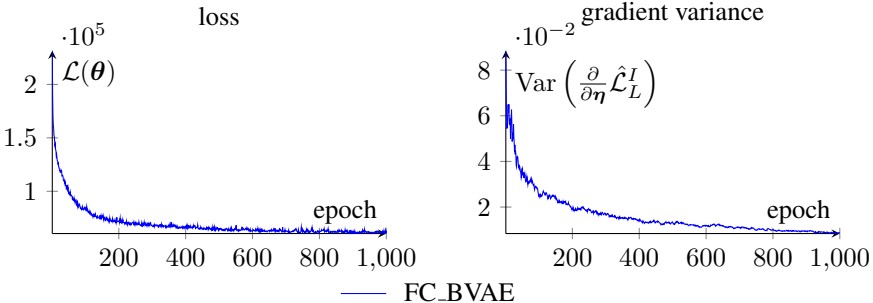

Figure 12: Convergence of the loss and gradient variance over 300 epochs of training on the fashion MNIST dataset.

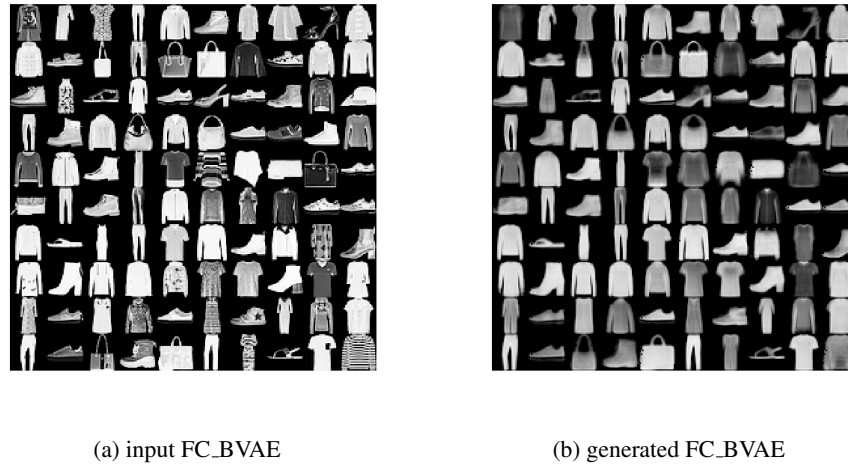

(a) input FC_BVAE          (b) generated FC_BVAE

Figure 13: Test input images and generated images of the FC_BVAE on the fashion MNIST dataset.

