# OpenReview forum: "Training Variational Auto Encoders with Discrete Latent Representations using Importance Sampling"
_ICLR.cc/2019/Conference_

### Official Review · AnonReviewer1 · 2018-10-23
**Rederivation of REINFORCE Estimator**

**Rating:** 3
**Confidence:** 5

**Review:**

Summary:
This paper proposes training VAEs with discrete latent variables by importance sampling the expected log likelihood (ELL) term in the ELBO, which is the problematic term since it is not amenable to reparametrization gradients.  For the importance sampling distribution, they choose the variational distribution itself, making the ELL gradient E[(d q(z|x) / d \theta) \log p(x|z) / q(z|x)].  Experiments are reported for MNIST and Fashion-MNIST using Bernoulli and categorical latent variables.

Critique:
The gradient estimator the paper proposes is the REINFORCE estimator [Williams, ML 1992] re-derived through importance sampling.  The equivalence can be seen just by expanding the derivative of log q in REINFORCE: E[log p(x|z) d log q(z|x)] = E[ (log p(x|z) / q(z|x)) d q(z|x) ], which is the exact estimator the paper proposes.  REINFORCE has been previously used for variational inference [Paisley et al., ICML 2012; Ranganath et al, AISTATS 2014] and deep generative models [Mnih & Gregor; ICML 2014] and recently extended for various control variates [Tucker et al., NIPS 2017].   The equivalence would not be exact if the authors chose the importance distribution to be different than the variational approximation q(z|x), so there still may be room for novelty in their proposal, but in the current draft only q(z|x) is considered.

Conclusion: Due to lack of novelty, I recommend rejection.


Miscellaneous points:
“...there exist no simple solutions to circumvent this problem.”  The Gumbel-softmax trick is fairly simple (although an approximation) [Jang et al., ICLR 2017; Maddison et al., ICLR 2017].

“...after training q(z|x) is a very good approximation to the true posterior p(z|x).”  That’s not necessarily true.

Equation #2 should be just equal to Equation #1.

“Kingma & Welling (2013) proposed to minimize L(\theta) using stochastic gradient descent on a training set...”. First uses of stochastic gradient for VI were [Sato, NC 2001; Platt et al., NIPS 2008; Hoffman et al., JMLR 2013].  Kingma & Welling [ICLR  2014] were the first to introduce reparameterized stochastic gradients.

Before Equation #11, the reference to Equation #4 should be to Equation #5.

“...the weighting...depends only on \theta_D and not on \theta_E” (p 4). D and E should be switched.

---

### Official Review · AnonReviewer3 · 2018-11-01
**It is a simple and trivial  extension of VAE with important sampling**

**Rating:** 1
**Confidence:** 5

**Review:**

This paper propose to use important sampling to optimize VAE with discrete latent variables. Basically, the methods proposed by this paper is rather simple and trivial. There are some discussions on why important sampling is not a good choice for VAE. Please refer: https://stats.stackexchange.com/q/255756

Moreover, if you focus on VAE with discrete latent variable, you should compare at least with Gumbel-Softmax: https://arxiv.org/abs/1611.01144

---

### Official Review · AnonReviewer2 · 2018-11-02
**A potentially nice idea that needs more thorough evaluation and discussion of related work**

**Rating:** 3
**Confidence:** 5

**Review:**

In computing the gradient of the ELBO, the main challenge lies in computing the gradient of the reconstruction loss with respect to the encoder parameters. VAEs traditionally rely on reparameterization in order to obtain a low-variance estimate, but there are a number of other gradient estimators that one can apply. The authors here proprose to use a trick that is known, but perhaps not widely known: If we introduce an importance sampling distribution, then we can use samples from this distribution to compute an importance-weighted estimate of the gradient. The idea is now that we can compute the gradient w.r.t. the encoder parameters as a simple importance-sampling estimate, which obviates then need for reparameterization, or likelihood-ratio estimators. The authors then apply this trick to train VAEs with discrete latent variables.

While I think that the idea that the authors present in this paper is worth further exploration, the paper in its current form is not sufficiently mature to appear at ICLR. The two areas where this paper would benefit from improvement are

1. Discussion of related work.

While the authors seem to suggest that there has been no work on VAEs with discrete latent variables, there has in fact been quite a lot of work, including work on VAEs that contain both discrete and continuous variables (e.g. [8-10], but I'm almost certainly missing further references). There has also been a large body of work on continuous relaxations of discrete variables that are amenable to reparameterization (e.g. [6-7], and references therein). There has also been a line of work relating importance sampling to variational objectives (see [1-3] as key references). Finally, there is also related work on reweighted-wake-sleep style objectives (see [4]) which similarly don't require reparameterization. From what I can tell, none of these references are cited or discussed as related work. In order to place this work in context, I would rewrite 2 to discuss approaches to gradient estimation in this space, which then makes it much easier to explain how this approach differs.

2. Empirical evaluation.

The authors only evaluate on MNIST and F-MNIST, and don't compare to any existing approaches. More than a couple of reconstructions, what I would like to see is an analysis of gradient variances, asymptotic ELBO estimates. I would also like to see a larger set of problems. Finally I would like to see a clear comparison to other methods based on, e.g., continuous relaxations.


References

[1] Y. Burda, R. Grosse, and R. Salakhutdinov, “Importance Weighted Autoencoders,” arXiv:1509.00519 [cs, stat], Sep. 2015.

[2] T. Rainforth et al., “Tighter Variational Bounds are Not Necessarily Better,” arXiv:1802.04537 [cs, stat], Feb. 2018.

[3] G. Tucker, D. Lawson, S. Gu, and C. J. Maddison, “Doubly Reparameterized Gradient Estimators for Monte Carlo Objectives,” arXiv:1810.04152 [cs, stat], Oct. 2018.

[4] T. A. Le, A. R. Kosiorek, N. Siddharth, Y. W. Teh, and F. Wood, “Revisiting Reweighted Wake-Sleep,” arXiv:1805.10469 [cs, stat], May 2018.

[5] A. Mnih and D. J. Rezende, “Variational inference for Monte Carlo objectives,” arXiv:1602.06725 [cs, stat], Feb. 2016.

[6] G. Tucker, A. Mnih, C. J. Maddison, J. Lawson, and J. Sohl-Dickstein, “REBAR: Low-variance, unbiased gradient estimates for discrete latent variable models,” in Advances in Neural Information Processing Systems, 2017, pp. 2624–2633.

[7] W. Grathwohl, D. Choi, Y. Wu, G. Roeder, and D. Duvenaud, “Backpropagation through the Void: Optimizing control variates for black-box gradient estimation,” arXiv preprint arXiv:1711.00123, 2017.

[8] J. T. Rolfe, “Discrete Variational Autoencoders,” arXiv:1609.02200 [cs, stat], Sep. 2016.

[9] E. Dupont, “Learning Disentangled Joint Continuous and Discrete Representations,” arXiv:1804.00104 [cs, stat], Mar. 2018.

[10] B. Esmaeili et al., “Structured Disentangled Representations,” arXiv:1804.02086 [cs, stat], Apr. 2018.

---

### Meta-Review · Area_Chair1 · 2018-12-01
**Missing important references and the proposed algorithm is essentially the well-known REINFORCE estimator**

**Confidence:** 5
**Recommendation:** Reject

**Metareview:**

The paper is addressing an important problem, but misses many related references (see Reviewer 2's comments for a long list of highly relevant papers).

More importantly, as Reviewer 3 pointed out (which the AC fully agrees):

"The gradient estimator the paper proposes is the REINFORCE estimator [Williams, ML 1992] re-derived through importance sampling."

"The equivalence would not be exact if the authors chose the importance distribution to be different than the variational approximation q(z|x), so there still may be room for novelty in their proposal, but in the current draft only q(z|x) is considered."